# The Use of Virtual Tissue Constructs That Include Morphological Variability to Assess the Potential of Electrical Impedance Spectroscopy to Differentiate between Thyroid and Parathyroid Tissues during Surgery

**DOI:** 10.3390/s24072198

**Published:** 2024-03-29

**Authors:** Malwina Matella, Keith Hunter, Saba Balasubramanian, Dawn Walker

**Affiliations:** 1Department of Computer Science, University of Sheffield, Sheffield S1 4DP, UK; m.matella@sheffield.ac.uk; 2Insigneo Institute for In Silico Medicine, Sheffield S1 3JD, UK; 3Liverpool Head and Neck Centre, Molecular and Clinical Cancer Medicine, University of Liverpool, Liverpool L69 7TX, UK; keith.hunter@liverpool.ac.uk; 4Department of Oncology and Metabolism, Royal Hallamshire Hospital School of Medicine and Biomedical Sciences, University of Sheffield, Sheffield S10 2RX, UK; s.p.balasubramanian@sheffield.ac.uk

**Keywords:** thyroidectomy, finite element modelling, model sensitivity study, thyroid and parathyroid tissue discrimination

## Abstract

Electrical impedance spectroscopy (EIS) has been proposed as a promising noninvasive method to differentiate healthy thyroid from parathyroid tissues during thyroidectomy. However, previously reported similarities in the in vivo measured spectra of these tissues during a pilot study suggest that this separation may not be straightforward. We utilise computational modelling as a method to elucidate the distinguishing characteristics in the EIS signal and explore the features of the tissue that contribute to the observed electrical behaviour. Firstly, multiscale finite element models (or ‘virtual tissue constructs’) of thyroid and parathyroid tissues were developed and verified against in vivo tissue measurements. A global sensitivity analysis was performed to investigate the impact of physiological micro-, meso- and macroscale tissue morphological features of both tissue types on the computed macroscale EIS spectra and explore the separability of the two tissue types. Our results suggest that the presence of a surface fascia layer could obstruct tissue differentiation, but an analysis of the separability of simulated spectra without the surface fascia layer suggests that differentiation of the two tissue types should be possible if this layer is completely removed by the surgeon. Comprehensive in vivo measurements are required to fully determine the potential for EIS as a method in distinguishing between thyroid and parathyroid tissues.

## 1. Introduction

There were, approximately, over 7000 thyroid gland operations recorded annually in the United Kingdom (before the 2020 COVID-19 pandemic) [1], an operation in which hypoparathyroidism and hypocalcaemia can occur, caused by inadvertent damage or removal of parathyroid glands. According to a British Association of Endocrine and Thyroid Surgeons report from 2021, the incidence rates of short- and long-term hypocalcaemia following total thyroidectomy are approximately 18.3% and 6.0%, respectively [1]. Therefore, endocrine surgeons seek additional methods (instrumentation or imaging tools) to decrease the incidence of the aforementioned complications through the successful identification and preservation of the parathyroid glands during surgery. Currently, there are few methods of differentiating the healthy or diseased parathyroid glands from the adjacent tissues; these methods either exploit their intrinsic autofluoroscopy [2,3] or the fluorescence detected after the administration of exogenous agents [4]. However, surgeons mostly rely on their own experience to preserve the parathyroid glands during thyroidectomy.

Recently, electrical impedance spectroscopy (EIS) measurements have been considered as an alternative noninvasive and real-time method with the potential to enhance the identification of parathyroid glands. EIS is based upon the resistive and reactive properties of biological tissues due to the presence of free and bound charges. This permits the measurement of these properties in the form of electrical impedance, defined as the frequency-dependent opposition to the flow of alternating electrical current. In a tetrapolar EIS device, an example of which is the commercial ${\mathrm{ZedScan}}^{TM}$ device (Figure 1a), which was designed to detect cervical intraepithelial neoplasia, a small alternating current is applied to a driving electrode (*I_1_*) and flows through the superficial layer of the tissue to the ground electrode *(V_0_*) while the passive electrodes *(V_1_*, *V_2_*) capture the potential difference. The principle of measurement with a tetrapolar probe is depicted in Figure 1b,c. The resultant impedance is calculated according to Ohm’s law, as a ratio of the potential difference and the applied current. The measurement is repeated over a range of frequencies and permits the capture of a characteristic impedance spectrum, including a substantial fall in the real part of the impedance in the kHz region (known as the β dispersion) due to the polarisation of cell membranes in the kHz frequencies [5].

Previously, EIS has been used for many applications in the fields of biology and medicine, as a means of differentiating between various animal tissue types [7] or to track the condition of tissues, such as the myocardium during the occlusion of the left anterior descending artery [8] or lungs during bronchoscopy as an alternative to histological biopsy [9]. However, the most widespread use of EIS in medicine is in the detection of cancer or precancer, including in studies related to benign and malignant skin pathologies [10], cervical cancer [11], oral tissues [12], breast tissue [13], prostate [14] and lung [15] cancers. Furthermore, several studies investigated the effectiveness of EIS devices in identifying pathologies in thyroid tissue [16,17,18] or to distinguish parathyroid glands from the adjacent tissues [19,20].

In particular, a study by Hillary et al. [19] investigated the ${\mathrm{ZedScan}}^{TM}$ (Zilico Ltd., Manchester, UK) device in an in vivo study for its ability to differentiate healthy parathyroid glands from the adjacent tissues, such as thyroid, brown adipose tissue and muscle, as well as pathological parathyroid glands. Hillary et al. [19] presented results that not only demonstrate a significant overlap in spectra obtained from healthy and pathological parathyroid glands but also show substantial similarities between the healthy median in vivo measured thyroid and parathyroid spectra. The study has also reported a wide range of impedances for both glands—530.30 Ω for thyroid and 433.68 Ω for parathyroid—at the frequency of 76 Hz, which contributes to the EIS spectra overlap. The derived Receiver Operating Characteristic (ROC) curve reported in that study showed 76% sensitivity and 60% specificity in distinguishing thyroid and parathyroid glands based on the EIS in vivo measurements.

In a previous publication [21], we developed computational multiscale-level Finite Element (FE) models to predict the electrical impedance spectra that relate to the EIS measurement with a tetrapolar ${\mathrm{ZedScan}}^{TM}$ probe. In that study, the modelling pipeline was utilised to investigate the impact of the tissue micro-, meso- and macroscale features (geometrical characteristics and electrical material properties) on the simulated spectra indices to elucidate their role in the bulk electrical properties of both tissues. An investigation was performed through a local sensitivity study, involving one-at-a-time variation of selected model parameters. The results, despite showing clear differences in the baseline simulated curves, suggested that thyroid and parathyroid differentiation may be challenging, due to the considerable overlap in the range of spectra obtained.

This sensitivity investigation [21] revealed the importance of particular geometrical parameters whose variations impact the electrical impedance spectra; specifically, extracellular space (ECS) thickness on the microscale, follicle size on the mesoscale and fascia thickness were demonstrated to have a significant impact on the measured electrical properties. Focusing on the macroscale, fascia is a type of loose connective tissue layer which anatomically covers the thyroid and parathyroid glands. During the surgery and prior to identification and impedance measurements, the surgeon aims to remove the fascia layer; however, it is impossible to guarantee the extent of its removal. Given that our earlier results suggested high model sensitivity to the fascia thickness and its material properties, the presence of this structure could therefore potentially ‘contaminate’ the in vivo measurements, contributing to the overlap between measurements from the tissues and leading to difficulties in distinguishing between these two tissue types, as well as previously observed discrepancies between the computed and in vivo measured spectra of thyroid and parathyroid glands. The contamination of EIS spectra by the fascia layer would arise due to differences in the morphological structure, and hence electrical properties, between this compartment and the follicular thyroid and cellular parathyroid tissues. Fascia is a type of loose connective tissue consisting of unstructured fibers of collagen, elastin and reticular fibers which are embedded in the extracellular matrix, contrasting with the higher-cellular-density tissues below. The presence of a fascia layer could result in the applied electrical current flowing predominantly through this tissue, instead of the underlying thyroid and parathyroid glands. Therefore, even a thin fascia compartment covering the glands might influence the differences in EIS spectra of the thyroid and parathyroid and obstruct their separability.

In this paper, we further investigate the parathyroid and thyroid tissue separability based on the in vivo measured and computed electrical impedance spectra. The dataset computed for this study will contain the EIS spectra obtained through the global sensitivity analysis to firstly investigate the impact of the variation of the morphological parameters throughout the micro- to macroscale. Generation of a set of finite element meshes or ‘virtual tissue constructs’ allows us to simulate sets of virtual impedance spectra that account for the natural variability in human thyroid and parathyroid tissue morphological features. After being used as the basis of a global sensitivity analysis, this dataset will be repurposed to evaluate the potential to distinguish between these tissues based on their macroscale impedance spectra, along with a separability investigation of the in vivo measured dataset collected by Hillary et al. [19]. The thyroid and parathyroid tissue classification will be performed using various statistical analysis and machine learning approaches. Moreover, the hypothesis of the fascia compartment presence potentially obstructing tissue differentiation will be further investigated.

## 2. Materials and Methods

### 2.1. Multiscale Thyroid and Parathyroid Model

Computational thyroid and parathyroid models have been developed previously and the details are summarised in our former publication [21]. Briefly, finite element modelling methods have been exploited to investigate the impact of various tissue features from different spatial scales. The requirement for a multiscale computational modelling approach arises from the hierarchical structure of tissues and the need for the inclusion of the cell membranes in the model geometry. The capacitive properties of the latter result in a characteristic reduction in impedance in the kHz frequency range, known as the β dispersion. It would not be computationally feasible to include such thin structures (∼8 nm) in a model volume of the order of centimetres, which is required to simulate tissue measurement with a tetrapolar EIS device on a macroscale. Therefore, the developed thyroid and parathyroid computational models consist of two to three levels of complexity: from microscale (cell scale), through mesoscale (follicle scale—unique to thyroid tissue only), to macroscale (tissue scale). The results from the lower-scale models are transferred to higher-level models, in the form of effective electrical material properties assigned to specific compartments at the higher scales. Examples of the geometries from different scales of the multiscale model for thyroid and parathyroid tissue are visualised in Figure 2.

The initial results [21] obtained from these models, in the form of computed EIS spectra, showed agreement with the in vivo results collected and reported by Hillary et al. [19]. The dependence of the model results, on both geometrical and compartmental electrical properties, was previously investigated using a local one-at-a-time method, which can be considered adequate as a first assessment of the model sensitivity [21]. However, by solely investigating the mean parameter values and their extremities, the local model sensitivity assessment leaves most of the parameter space unexplored, as well as neglecting the non-linear and combinatorial parameters effects [22]. In this paper, the same modelling pipeline was used to perform a global sensitivity study, where many of the morphological model parameters have been varied and investigated simultaneously. Such an analysis permits a more thorough exploration of parameter space and the capture of potential interactions between parameters. Subsequently, virtual spectra, generated to account for variability in key parameters, were used for an analysis of tissue separability, based on Receiving Operating Characteristic (ROC) curves, and finally, a set of machine learning models were assessed for their ability to separate the simulated spectra for thyroid and parathyroid tissue.

### 2.2. Global Sensitivity Analysis

#### 2.2.1. Model Parameters

The preliminary local sensitivity study results [21] permitted the number of parameters, and in some instances their range, to be narrowed down prior to the global sensitivity study presented in this paper. Overall, there were six and five geometrical parameters considered in the global sensitivity analysis for thyroid and parathyroid models, respectively. Due to the positive agreement between the baseline curves obtained with the default model parameter values with the in vivo measured EIS range presented in the previous study [21], the electrical material properties investigation was excluded from this study, and electrical conductivity σ and relative permittivity $\varepsilon_{r}$ were fixed to their baseline values, which are summarised in Appendix A, Table A1.

The geometrical parameter values were selected based on the values reported in the literature, manually measured using the image analysis software (ImageJ (U. S. National Institutes of Health, Bethesda, MD, USA) and Aperio ImageScope (Leica Biosystems Imaging, Nussloch, Germany)) or estimated. The model input parameters investigated in the global sensitivity analysis, along with the information on their distribution and their distribution indices (minimum and maximum values for uniform distribution and mean and standard deviation values for normal distribution), are summarised in Table 1 for thyroid, and Table 2 for parathyroid tissue.

In order to further explore the effects of the fascia layer on the EIS spectra and to assess its impact on the thyroid and parathyroid tissue differentiation, the geometrical parameter sensitivity analysis was performed twice for each gland: once including and once excluding this superficial compartment. In summary, the global sensitivity analysis comprises four separate sub-studies, where, for each gland, two sets of model evaluations have been performed and these are (i) geometrical parameters including the fascia compartment, (ii) geometrical parameters excluding the fascia compartment.

#### 2.2.2. Parameter Space Sampling

Parameter space sampling was performed using the Latin Hypercube Sampling (LHS) method. Considering the significant simulation time for solving each multiscale model (30–90 min for each multiscale model simulation to solve at 14 frequencies), the LHS was the preferred sampling method over the standard Monte Carlo approach, due to the lower number of model evaluations required for in the LHS approach [27]. In the LHS method, the samples from the parameter range are generated based on their probability density function, by dividing it into N (number of samples) non-overlapping and equiprobable intervals. From each interval, a sample is selected randomly without replacement. The parameter sampling was performed using the pyDEO Design for Experiments for Python open-source package. An initial study required to establish the optimal sample size for the analysis has been described in [23]. Due to the heavy computational load, it was decided to maintain a constant sample size of 100 model simulations per sensitivity analysis sub-study.

#### 2.2.3. Sensitivity Assessment

There are two general approaches to model global sensitivity assessment: the variance decomposition and correlation-based methods [28]. The first group includes the comprehensive Sobol method [29], which is the preferred approach when investigating the contribution of each model input parameter to the model output variance. This method also accounts for the effect of parameter interaction through the higher-order indices. Due to the requirement of a significant number of model evaluations (in the order of a few thousand), the Sobol method was not implemented in this study. Instead, the correlation-based method was the preferred option to assess the sensitivity of the multiscale thyroid and parathyroid models. In this approach, the model sensitivity is presented as the strength of the linear correlation between model inputs and outputs. For non-linear models, the Partial Rank Correlation Coefficient (PRCC) is an approach where the input–output association is determined based on their rank-transformed values. The PRCC calculation and its comparison to the remaining correlation coefficients is summarised in Marino et al. [28].

The PRCC ranges from −1 to 1, where the extremities represent the highest linear association between an input–output pair. This association decreases when the PRCC approaches zero, which indicates no linear association. Positive PRCC indicates that the output parameter values increase with the increase in the input, while negative PRCC suggests an inverse proportional relationship. The coefficient’s range can be divided into three groups that determine the association strength between parameters: low (<0.4), medium (>0.4 and <0.7), and high (>0.7) levels. In this study, the PRCC for each input and output pair was calculated and then assigned to the relevant association strength group.

#### 2.2.4. EIS Spectra Parameterisation

Similarly to our previous approach [21], simulated spectra were parameterised by choosing three output indices: $Z_{1}$ and $Z_{14}$, which are two impedance values at the first and last simulated frequency points (76 Hz and 625 kHz), relating to the impedance values before and after β dispersion, and the frequency $f_{mid}$ at the middle of the dispersion (the frequency at which impedance takes the middle value between $Z_{1}$ and $Z_{14}$). These parameters were used to assess the model sensitivity and as thyroid and parathyroid spectra separability indices.

## 3. Model Verification and Differentiation Assessment

The computed results were compared with the in vivo measurements that were previously published by Hilary et al. [19] and the data were used in this study with the authors’ permission. Briefly, the existing in vivo measured dataset comprises 53 thyroid and 42 parathyroid EIS spectra which were acquired during different types of thyroid and parathyroid surgeries with the tetrapolar ${\mathrm{ZedScan}}^{TM}$ device. For the purposes of qualitative comparison, the in vivo measured spectra were parameterised similarly to the computed results, by assigning the three impedance spectra indices: $Z_{1}$, $Z_{14}$ and $f_{mid}$.

The two computed (including and excluding the fascia compartment) and in vivo measured datasets were used to assess the feasibility of distinguishing between thyroid and parathyroid tissue, based on the selected spectra indices ($Z_{1}$, $Z_{14}$ and $f_{mid}$). The potential for tissue differentiation was then evaluated using three approaches. Firstly, thyroid and parathyroid data were compared qualitatively using 2D scatter plots for all spectra indices combinations. Secondly, tissue differentiation of each computed and in vivo measured dataset was assessed by manually plotting the ROC curves and calculating the Area Under Curve (AUC) for each spectra index, as a classification determinant, separately. For each dataset and spectra index, the true positive (TP), true negative (TN), false positive (FP) and false negative (FN) values were calculated while increasing the thyroid/parathyroid classification threshold. These parameters were utilised to derive the True Positive Rate (TPR) and False Positive Rate (FPR) from the following Equations (1) and (2):(1)TPR=TP/(TP+FN),
(2)FPR=FP/(FP+TN).

TPR and FPR values for each threshold are used to plot the ROC curves, which subsequently permits the calculation of the AUC, which is an indicator determining the effectiveness of a classification. AUC can be interpreted as the probability of correct positive case classification between a randomly selected pair of negative and positive cases [30]. AUC values of 0.5 or lower suggest that the model’s performance is equal to or worse than a random classification, while values close to one suggest a perfect classification.

Finally, all three spectra indices were considered as features in a classification study using three approaches: Support Vector Machine (SVM), K-Nearest Neighbour (KNN) and Random Forest Classifier (RFC). All three classifiers are examples of supervised machine learning models, and have been selected based on their successful implementation in electrical impedance signal binary classification, as documented in the literature [15,31]. Full details of the principles of each classifier can be found in the scikit-learn documentation [32]. Briefly, SVM aims to identify the optimal surface or hyperplane separating the data by the most substantial margin, where the distance between the data points from different classes is maximised. A KNN classification is performed based on the closest proximity of a new data point to the labeled data previously used in training. By contrast, the RFC is an ensemble method, which is built upon the principle of creating multiple decision trees which combine their prediction to perform the classification. Each decision tree is based on multiple conditional statements referring to the input features, in order to assign objects into one of the classes. The performance of the classifiers was assessed using the metrics of mean accuracy and AUC from 20-fold cross-validation. Classification using the supervised learning methods was applied only to the computed data, due to the small size of the in vivo measured EIS dataset. This part of the study was performed using scikit-learn, the open-source data analysis and machine learning Python library [32].

## 4. Results

### 4.1. EIS Spectra Computation

The computational study resulted in 400 thyroid and parathyroid multiscale simulated impedance spectra; two sets of simulations, including or excluding the fascia compartment, were performed for each gland, respectively. Each simulation was performed at 14 frequencies that correspond to the frequencies used in the in vivo measurements with the ${\mathrm{ZedScan}}^{TM}$ tetrapolar probe [19]. The resultant impedance spectra, for each set of input parameters, were evaluated qualitatively by plotting all impedance values against 14 frequencies from 76 to 625,000 Hz. All impedance values presented in this paper show the frequency-dependent real component of impedance values computed in each simulation. For the purposes of direct comparison with the pre-existing in vivo dataset [19], only the real part of the impedance from the computed dataset will be evaluated in this study. The computed data were used to assess the global model sensitivity and to verify the computational results against the in vivo measured EIS spectra. The simulated datasets, which were reflective of the expected range of spectra arising from the natural variability of the selected morphological features of the tissues, were then repurposed to evaluate the potential for thyroid and parathyroid tissue separation based on the selected spectra indices.

### 4.2. Global Sensitivity Macroscale Results

The PRCC results of the global sensitivity analysis at the macroscale level, representing the correlation between the input parameters and selected spectra indices ($Z_{1}$, $Z_{14}$ and $f_{mid}$), are presented in Table 3 and Table 4 and the outcomes are divided by gland and the inclusion of the fascia compartment in the model. The micro- and mescoscale global sensitivity results are summarised in Appendix B, Table A2 and Table A3. The PRCC indices included in Table 3 and Table 4 are characterised with the probability parameter *p* < 0.001. Most of the parameters have a low-level association (PRCC < 0.3), and these have not been included among the sensitivity study results presented in this section.

#### 4.2.1. Thyroid Results

Inspection of Table 3 reveals that the high-frequency impedance index $Z_{14}$ from the simulation set, including a fascia compartment, exhibits high sensitivity to fascia thickness. By contrast, the low-frequency impedance $Z_{1}$ is correlated with the size of the follicle ($d_{follicle}$) and ECS thickness ($d_{ECS}$), while the mid-dispersion frequency ($f_{mid}$) also exhibits a high association with the latter.

The results from the macroscale model simulations excluding the fascia compartment, reveal a greater impact of the lower-scale parameters, compared to the results from the equivalent model including fascia. This is especially noticeable for $Z_{14}$, where a high association with the size of the follicle ($d_{follicle}$) can be observed, which is not the case for the model including fascia. Moreover, a high association can also be noted between $Z_{1}$ and input parameters such as $d_{ECS}$ and $d_{follicle}$, and between $f_{mid}$ and $d_{ECS}$.

#### 4.2.2. Parathyroid Results

In the parathyroid macroscale model including the superficial fascia layer, this compartment continues to be an influential parameter, showing a high association with $Z_{14}$ and $f_{mid}$ spectra indices. Furthermore, similarly to the thyroid results, $d_{ECS}$ is a crucial parameter at the parathyroid macroscale, showing a high and medium association with $Z_{1}$ and $f_{mid}$ spectra indices, respectively.

The results computed using the macroscale model without the inclusion of fascia revealed that $y_{cell}$ and $d_{ECS}$ have stronger association with the $Z_{1}$ and $Z_{14}$ parameters, compared to the parathyroid results from the simulations including the superficial layer. Moreover, the $f_{mid}$ index is also impacted by these two parameters, with a PRCC suggesting a medium association.

### 4.3. Computed Spectra Verification against Measured Data

The computed impedance spectra obtained through the global sensitivity analysis based on the investigated group of the input parameters for both glands, with and without fascia, are presented in Figure 3 in comparison to the range of in vivo measured spectra [19]. The red spectra marked on all plots signify the baseline results obtained with the default parameter values from the previous investigation [21]. The computed results obtained through the variation of the geometrical parameters including the fascia compartment, plotted in Figure 3a,b lie within or close to the range of the in vivo measured spectra for both glands. The best agreement between the computed and measured data is especially noticeable at frequencies below 100 kHz, while at higher frequencies, the computed results tend to under-predict the impedance for both glands when compared with the in vivo experimental results.

A better agreement in this high-frequency region can, however, be observed in Figure 3c,d, where the impedance spectra have been acquired through the variation of geometrical parameters excluding the superficial fascia compartment. Nonetheless, in the frequency region below 100 kHz, a small number of the simulated thyroid and parathyroid impedance spectra from this group of results fall above the upper limits of the in vivo measured results.

### 4.4. Thyroid and Parathyroid Tissue Differentiation

#### 4.4.1. Qualitative Separation

The separability of the thyroid and parathyroid spectra was assessed qualitatively using 2D scatter plots to visualise the relationship between each pair of selected impedance spectra indices. The scatter plots are shown in Figure 4 for all investigated datasets: in vivo measured (Figure 4a–c), computed including (Figure 4d–f) and excluding (Figure 4g–i) fascia compartment.

Inspection of Figure 4 reveals the best visual separation of the thyroid and parathyroid data is obtained for simulated data without the fascia compartment (Figure 4g–i) based on all pairs of spectra indices. All three plots show non-overlapping clusters of thyroid and parathyroid results, implying a complete separation of the results of these tissue types. The in vivo measured and computed results including fascia (Figure 4a–f) suggest more difficulty in separating thyroid and parathyroid results due to the notable overlap between the data in all the spectra index configurations. In these cases, however, the best qualitative separation of the thyroid and parathyroid results is shown in the $f_{mid}$ against $Z_{1}$ plot.

#### 4.4.2. Manual ROC-Based Classification

The results of the thyroid and parathyroid manual classification based on the individual spectra indices ($Z_{1}$, $Z_{14}$, $f_{mid}$), which was performed for all three datasets (in vivo measured, computed including and excluding the fascia compartment), are visualised in Figure 5 in the form of ROC curves. Moreover, Table 5 summarises the quantitative results of the calculated AUC with regards to the investigated dataset and the selected spectra index, based on which the thyroid/parathyroid classification was performed. The results presented in this section show the more favourable outcome for thyroid and parathyroid separation after testing two instances: a given parameter being higher for thyroid or for parathyroid.

Confirming the initial observations from the qualitative comparison (Figure 4), the best data separation is noted in the computationally derived dataset without the fascia compartment (Figure 5a). For this dataset, two parameters, $Z_{14}$ and $f_{mid}$, provide AUCs over 0.90, recommending these parameters as good thyroid and parathyroid separation indices. The particularly high value of 0.998 for $Z_{14}$ implies an almost perfect classification of the simulated results in the case where there is no superficial fascia.

Figure 5b and Table 5 show how the inclusion of fascia in the computational model changes the tissue separability, indicating that this compartment lowers the AUC scores for the selected spectra indices $Z_{14}$ and $f_{mid}$. However, the separability based on the low-frequency impedance ($Z_{1}$) remains at a comparable level regardless of the fascia inclusion (AUC of 0.732 and 0.721 for the dataset including and excluding fascia respectively). Moreover, the inclusion of the fascia not only lowers the ability to distinguish the tissues based on $Z_{14}$, but also suggests the high-frequency impedance of thyroid should be higher compared to parathyroid, which contradicts the results from models without fascia, as shown in Table 5.

Inspection of Figure 5c and Table 5 reveals lower separability of the in vivo measured thyroid and parathyroid results in comparison to the computed results from the model excluding fascia. Moreover, there are discrepancies in the $Z_{1}$ parameter, which suggest that low-frequency impedance is higher for thyroid than parathyroid in the in vivo measured results, which is in contradiction with the outcomes of both computational studies. Nonetheless, comparably high AUC values for all the investigated datasets are observed for the dispersion frequency parameter ($f_{mid}$ AUC = 0.862 from the in vivo measured dataset), suggesting that this index could be the optimal candidate for the basis of thyroid and parathyroid impedance spectra separability.

#### 4.4.3. Machine Learning Classifier Analysis

Finally, thyroid and parathyroid tissue separability was investigated using three supervised learning classification algorithms, trained on all three selected spectra indices derived from the two computed datasets only. This assessment was not carried out for the in vivo measured dataset due to insufficient data. The selected models used for classification were the Support Vector Machine, K-Nearest Neighbour and Random Forest Classifier. The summary of the classifiers’ performance based on the mean AUCs and accuracies from the 20-fold cross-validation is presented in Table 6. The thyroid and parathyroid data separation from the model simulations excluding the fascia compartment was successful using all three classifiers, where the results show AUC and accuracy values over 0.90 (with the exception of the accuracy of KNN—0.879). The inclusion of fascia significantly lowered the performance of the SVM and KNN classifiers; Table 6 shows the reduction in both AUC (from 0.978 to 0.649 for SVM and from 0.956 to 0.608 for KNN) and accuracy (from 0.908 to 0.588 for SVM and from 0.879 to 0.517 for KNN). RFC demonstrated the most favourable performance in classifying the thyroid and parathyroid tissue from both investigated datasets, where the fascia inclusion in the model showed a smaller decrease in the tissue separability, in comparison to the remaining classifiers (AUC change from 1.000 to 0.918 and accuracy change from 0.994 to 0.840).

## 5. Discussion

The purpose of this study was to re-evaluate the healthy thyroid and parathyroid tissue separability based on computational analysis of both simulated and pre-existing in vivo electrical impedance spectroscopy data. The simulated data, generated using the previously developed multiscale models of thyroid and parathyroid tissue [21], allowed a global sensitivity analysis which explored the impact of the natural variation of the geometrical properties of tissue features on the characteristics of the impedance spectra. A separability investigation was then performed on the pre-existing measured data, as well as spectra generated from computational models in two configurations: with and without the macroscale superficial fascia compartment, which was hypothesised to have a negative effect on the ability to distinguish between thyroid and parathyroid tissue. Moreover, thyroid and parathyroid separation was evaluated through a qualitative and quantitative comparison, which included statistical and machine learning methods for classification of thyroid and parathyroid based on the selected spectra indices.

Virtual EIS spectra were simulated through the variation of the geometrical parameters defining the finite element meshes at different scales, which fulfilled a dual purpose: firstly, as part of a global sensitivity investigation and secondly, as a “virtual EIS dataset”—considered representative of the range of electrical impedance spectra anticipated as a result of natural variability in thyroid and parathyroid tissue morphology expected within a set of virtual patients. The latter dataset was then analysed for tissue separability, based on extracted features of the virtual EIS curves—low- and high-frequency impedance, and the mid-dispersion frequency. The global sensitivity results for the computed datasets including the fascia compartment, presented in Section 4.2, revealed high correlation between selected spectra indices and the ECS layer thickness and fascia layer thickness for both tissues, which is in agreement with the initial outcomes from the local sensitivity analysis [21]. Moreover, the high correlation between fascia thickness and the high-frequency impedance ($Z_{14}$) for both tissues and the dispersion frequency ($f_{mid}$) for parathyroid tissue further confirms the significant influence of the fascia compartment on the computed EIS spectra. By contrast, the exclusion of this compartment in the parallel global sensitivity analysis resulted in increased influence of the cell size ($y_{cell}$) and follicle size ($d_{follicle}$) on the selected spectra indices for parathyroid and thyroid tissue, respectively. Note that these morphological parameters are characteristic and unique to each tissue type.

In future work, it will be crucial to further consider the sample size of simulated spectra considered in the global sensitivity analysis. In this study, the sample size was set to 100 model evaluations after an initial study [23], which suggested the correlation level between given model input–output remains the same with increasing sample size. However, it is important to bear in mind that most of the parameters did not achieve convergence in their PRCC values. Considering the high computational load of the multiscale simulation, surrogate modelling methods could be implemented in order to run a larger set of simulations, leading to convergence in the sensitivity indices and a more reliable quantitative sensitivity assessment. Surrogate modelling methods have been successfully implemented in the past in the sensitivity studies investigating cardiac cell electrophysiology [33] or four chamber heart hemodynamics model [34] and could be incorporated into future model sensitivity analysis in the context of EIS.

Figure 3 visualises the simulated spectra against the in vivo measured spectra, revealing that the computed results obtained with models including and excluding the fascia compartment fall within, or are close to, the in vivo measured range. The best match is observed in the low-frequency region (below 100 kHz) for both datasets. In the high-frequency region, beyond the β dispersion range (over 100 kHz), there is an improved agreement within the results of the model without fascia. The initial under-prediction of the impedance seen in the results including the fascia compartment could be explained by the structure and electrical properties of fascia—specifically, its higher overall conductivity when compared to the cellular parathyroid and follicular thyroid structures, which is manifested mostly in the high-frequency region. Nonetheless, a small number of simulated impedance spectra in the range between 10 kHz and 100 kHz exceed the in vivo measured impedance range, which could be explained by the homogenisation assumption inherent in our multiscale modelling approach. Specifically, as discussed in [35], discarding the heterogenous nature of biological tissues in theoretical modelling could result in narrow β dispersion and higher dispersion frequencies in comparison to the in vivo measured spectra. However, the overall positive agreement between the computed and in vivo measured spectra suggests that the chosen modelling methods are suitable and can generate realistic simulated EIS spectra for thyroid and parathyroid tissue which can justifiably be used in our subsequent classification study.

All three methods of thyroid and parathyroid tissue separability assessment revealed similar outcomes, showing the best separability between the tissues based on the computed dataset excluding the fascia compartment. The qualitative comparison from Figure 4g–i (without fascia) showed clear separation of thyroid and parathyroid tissues based on all three spectra index combinations. However, the in vivo measured and the remaining simulated dataset (including fascia) did not exhibit clear thyroid and parathyroid separability, showing a significant overlap between the selected spectra indices in these cases. Nonetheless, for the in vivo measured dataset and computed dataset including fascia, the model demonstrated the best qualitative separation of the thyroid and parathyroid results on the basis of comparing $Z_{1}$ and $f_{mid}$ parameters.

The manual classification results, using the individual spectra parameters as classification indices, are shown in the form of ROC curves and derived AUC values in Figure 5 and Table 5. Similarly, the tissue classification was the most successful for the simulated dataset excluding the fascia compartment. The AUC values for all spectra indices were higher than 0.70 and, for the high-frequency impedance ($Z_{14}$), reached 0.998, implying an almost perfect classification. This result, however, is not observed in the in vivo measurements or simulated spectra including fascia. It is further demonstrated by the ROC curves and AUC results that the inclusion of fascia in the computational model lowers the separability of thyroid and parathyroid tissue, especially using the $Z_{14}$ and $f_{mid}$ indices (0.527 compared to 0.998 for $Z_{14}$ and 0.644 compared to 0.971 for $f_{mid}$). Similarly, there is a lower AUC value reported for thyroid and parathyroid differentiation in the in vivo dataset. These results suggest that these in vivo tissues could have been covered by a thin layer of the connective tissue during EIS measurement, which contaminated the spectra and hence decreased the separation of these two glands. Overall, considering all three investigated datasets, the best separability was demonstrated for the $f_{mid}$ index, recommending the frequency in the middle of the dispersion as a promising discriminant of the healthy thyroid and parathyroid tissues.

Finally, three machine learning classifiers were implemented to further evaluate the potential for separation of thyroid and parathyroid tissues based on all three spectra indices obtained from the simulated data. The performance of the classifiers was evaluated for two computed datasets, including and excluding fascia compartment. All three classifiers demonstrated a good performance in distinguishing between the tissue types in the dataset generated by models without fascia, with AUCs over 94%, which is consistent with the findings of the earlier separability evaluations. Also, similarly to the outcomes of the manual classification using ROC curves, the performance of SVM and KNN decreases when applied to the spectra derived from the model including fascia. Nonetheless, the ensemble method, RFC classifier, successfully separated this dataset with the AUC of 0.918 and accuracy of 0.840. Since the RFC showed the best performance, it would be recommended as the classifier of choice in differentiating between the parathyroid and thyroid tissues for the two computed datasets.

The utility of machine learning classification methods in recognising different tissue types has already been demonstrated in the literature; for instance, SVM, KNN and linear discriminant analysis models were successful in identifying pulmonary nodules [15] with prior principal component analysis, which resulted in AUCs higher than 90% and accuracies over 95%. In another study, two neural network models were implemented in classifying healthy and cancerous breast tissue with classification accuracies over 93% [36]. A machine learning approach has also been implemented to identify preterm birth based on EIS measurements from the cervix, with the most favourable separability results of AUC of 0.80 reported for women before undergoing any medical treatment [37]. The first two studies reported classification results slightly more favourable than these reported in this study using machine learning methods. It is difficult to directly compare these results, however, due to the differences in types of data used for classification and different methodologies; nonetheless, there may be potential to increase the performance of the classifiers used in this study through data preprocessing, normalisation and appropriate feature selection, which were not performed here. In addition, future studies could explore thyroid and parathyroid separability based on a different set of spectra parameters, e.g., using another subset from the 14 computed/measured impedance values, the imaginary part of impedance, the phase difference or the Cole parameters, which are frequently used in studies investigating electrical properties of tissues to represent the impedance spectra features. For example, various combinations of Cole parameters have been investigated and successfully used as the features to distinguish cervical intraepithelial neoplasia from healthy tissues using machine learning classifiers [38]. The Cole model fitting methods were not applied in this study, due to the limitations of the shape of parathyroid spectra and the frequency range in the experimental dataset, as discussed in [23].

As shown in Figure 5 and Table 5, the inclusion of a fascia compartment in the model did not affect the tissue separability based on the low-frequency impedance parameter ($Z_{1}$), suggesting that this index is less sensitive to the presence of the connective tissue than the other spectra parameters investigated in this study. This lack of sensitivity was also demonstrated in the global sensitivity study results. Therefore, the reasons for the observed discrepancies in $Z_{1}$ index (the in vivo data classification result suggests higher low-frequency impedance for thyroid over parathyroid, with the opposite trend demonstrated by the computed datasets) cannot be attributed to the potential presence of fascia. Discrepancies in the low-frequency impedance could be explained by factors which were not accounted for in the computational model of thyroid and parathyroid tissue, or by the significant range and overlap in the thyroid and parathyroid in vivo measured low-frequency impedance. For instance, the presence of different tissue structures or pathologies [19], the effects of temperature [39], gland activity [40], viability [19] and hydration [41] are all factors which could have had an impact on the discrepancies observed between the in vivo measured and computed results. Moreover, uncertainties in the electrical material properties of various cellular and tissue components, which were not investigated in this study, could have added to the observed discrepancies, as discussed in a previous publication [21]. Finally, a previous investigation of the accuracy in the EIS measurements on small structures, such as the parathyroid glands, acquired with the tetrapolar ${\mathrm{ZedScan}}^{TM}$ probe, revealed that probe/parathyroid misalignment could significantly reduce this gland’s impedance in the low-frequency region when compared to a symmetrical and precise measurement [42], which could also potentially explain the observed discrepancies.

It was not possible to assess the performance of the machine learning classifiers in separating the in vivo measured spectra, due to the insufficient number of measurements (53 thyroid and 42 parathyroid EIS spectra). To further verify the outcomes of the classification study presented in this paper, it will be crucial to perform further, more comprehensive in vivo measurements. Despite the initial positive agreement of many aspects of the computed results with the measurements presented here, access to additional in vivo data, ideally with supplementary tissue information, e.g., actual geometrical parameters based on histology images, would permit further model validation, confirming the reliability of our simulation approach. Bearing in mind the ethical considerations to minimise the need of invasive procedures required to access healthy patient tissues in vivo, future model validation work could be focused on tissue engineered models. Moreover, the development of tissue engineered samples for EIS measurements would permit the collection of real geometrical tissue features without the need to perform an invasive biopsy to obtain the necessary histology data. Finally, future in vivo and computational modelling work could be extended to other tissue types in close proximity to the thyroid/parathyroid glands, such as brown fat or lymph nodes, which would broaden our ability to differentiate between these tissue types.

## 6. Conclusions

In summary, in this study, we assessed thyroid and parathyroid tissue separability based on selected electrical impedance spectra indices extracted from both measured in vivo and simulated impedance spectra, generated by a set of virtual tissue models. The results of a global sensitivity analysis and separability study demonstrated the high sensitivity of the thyroid and parathyroid impedance spectra to the superficial fascia layer, showing that its presence lowers tissue separability based on both manual and machine learning classification methods. Assuming that this surface tissue layer could be completely removed by the surgeon, our manual classification (ROC-based) study suggests that positive AUC values of over 0.90 could be obtained, based on extracting the high-frequency impedance ($Z_{14}$) and the mid-dispersion frequency ($f_{mid}$) indices, recommending them as potential discriminants for thyroid and parathyroid tissue separability. From the machine learning classifiers investigated, the Random Forest Classifier demonstrates the best performance in distinguishing between the thyroid and parathyroid glands, even when considering the presence of fascia (AUC of 0.918 after fascia inclusion). Further in vivo measurements are required to verify the outcomes of this study, and to find the optimal approach to the most effective thyroid and parathyroid tissue separability during thyroidectomy.

## Figures and Tables

**Figure 1 sensors-24-02198-f001:**
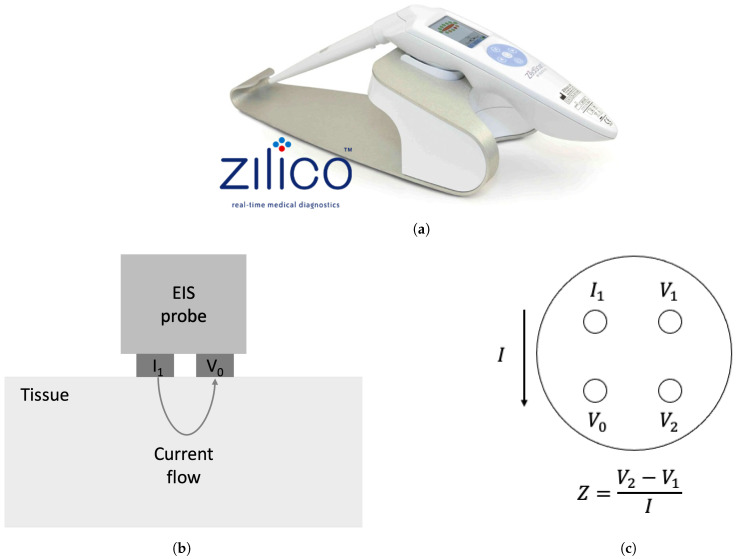
Electrical impedance spectroscopy device: (**a**) ZedScan™ [6], (**b**) the schematic showing the current injected from EIS probe and flowing through the tissue, (**c**) tip of the tetrapolar probe showing the principle of the impedance *Z* measurement; a known current *I* flows between the active electrodes (*I_1_* and *V_0_*) while the passive electrodes (*V_1_* and *V_2_*) capture the potential difference at each frequency.

**Figure 2 sensors-24-02198-f002:**
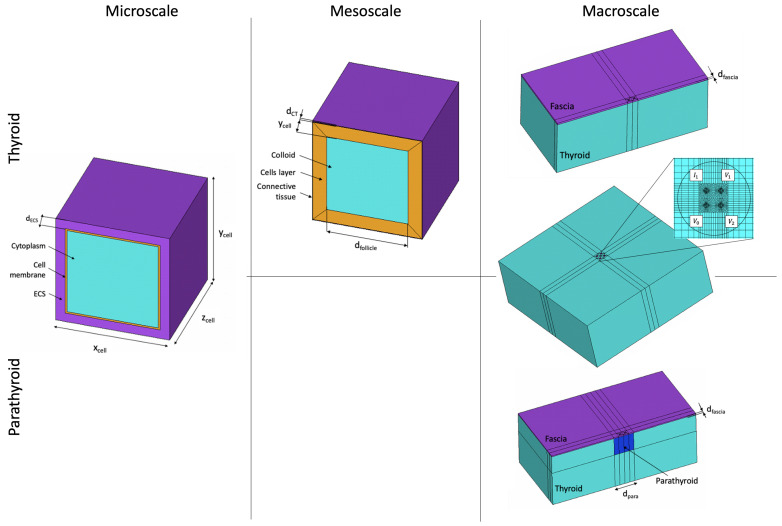
Thyroid and parathyroid model geometries across the multiscale pipeline with marked geometrical features: $x_{cell}$, $y_{cell}$, $z_{cell}$—cell dimensions, $d_{ECS}$—extracellular space thickness, $d_{follicle}$—size of follicle, $d_{CT}$—follicular connective tissue thickness, $d_{fascia}$—fascia thickness, $d_{para}$—size of the parathyroid gland, $I_{1}$—driving electrode, $V_{0}$—ground electrode, $V_{1}$ and $V_{2}$—passive electrodes.

**Figure 3 sensors-24-02198-f003:**
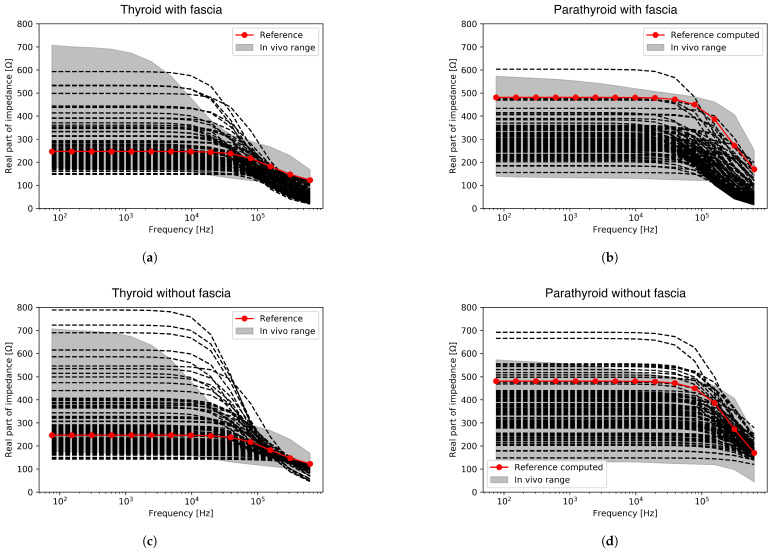
Comparison of the computed spectra (black dashed lines, with the red dotted line marking the baseline spectrum) against the range of experimental data (grey range) for (**a**,**c**) thyroid, (**b**,**d**) parathyroid tissue investigations.

**Figure 4 sensors-24-02198-f004:**
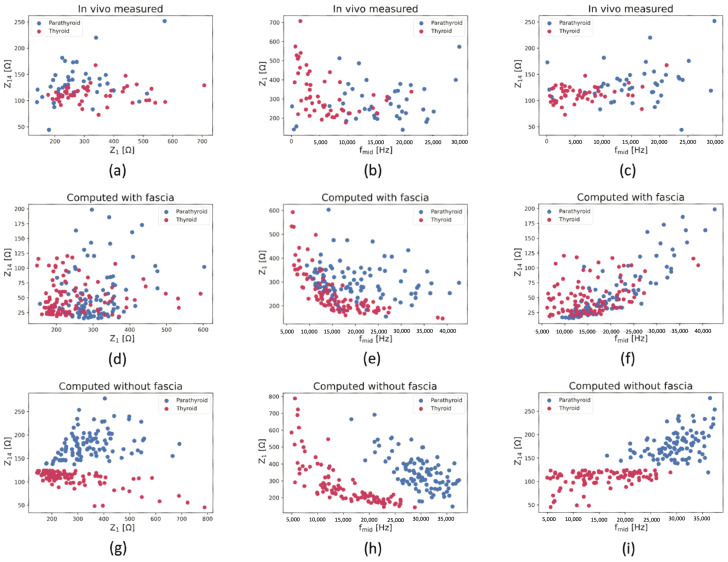
Scatter plots of the selected spectra indices visualising the spread of the computed and in vivo experimental thyroid and parathyroid results: (**a**–**c**) in vivo measured results, (**d**–**f**) computed results including fascia, (**g**–**i**) computed results excluding fascia; thyroid results are marked with red markers, parathyroid with blue.

**Figure 5 sensors-24-02198-f005:**
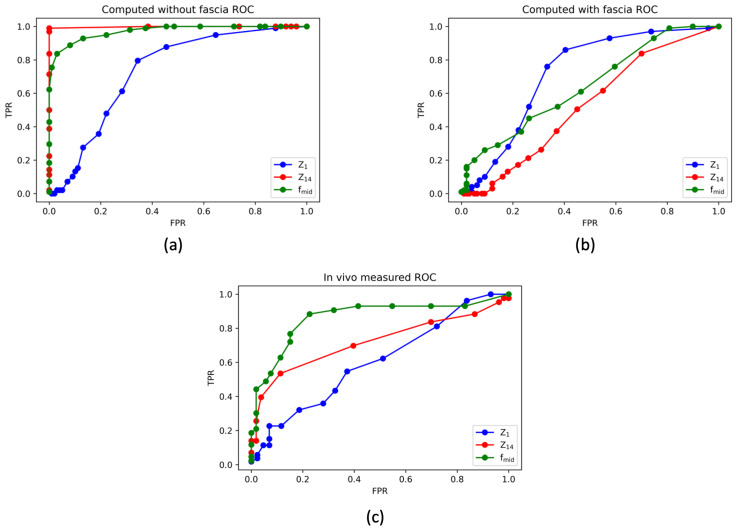
ROC curves for the thyroid and parathyroid classification based on individual selected spectra indices ($Z_{1}$—impedance at 76Hz, $Z_{14}$—impedance at 625kHz, $f_{mid}$—frequency in the middle of the dispersion) for three datasets: (**a**) computed excluding fascia, (**b**) computed including fascia, (**c**) in vivo measured.

**Table 1 sensors-24-02198-t001:** Thyroid tissue model input geometrical parameters and their probability distribution information: $x_{cell}$, $y_{cell}$—cell dimensions, $d_{ECS}$—extracellular space thickness, $d_{follicle}$—size of follicle, $d_{CT}$—follicular connective tissue thickness, $d_{fascia}$—fascia thickness.

Parameter	Distribution	Distribution Indices	Value	Reference
Microscale				
$x_{cell}$ [µm]	Normal	Mean Standard deviation	8.53 µm 1.84 µm	Histology measurements [23]
$y_{cell}$ [µm]	Normal	Mean Standard deviation	8.53 µm 1.84 µm	Histology measurements [23]
$d_{ECS}$ [µm]	Uniform	Min Max	0.1 µm 0.5 µm	Initial local sensitivity results and estimated from cervical epithelium measurements [24,25]
Mesoscale				
$d_{follicle}$ [µm]	Normal	Mean Standard deviation	113.77 µm 63.13 µm	Histology measurements [23]
$d_{CT}$ [µm]	Uniform	Min Max	0.8 µm 2.5 µm	Estimated
Macroscale				
$d_{fascia}$ [mm]	Uniform	Min Max	0.025 mm 0.5 mm	Estimated

**Table 2 sensors-24-02198-t002:** Parathyroid tissue model input geometrical parameters and their probability distribution information: $x_{cell}$, $y_{cell}$—cell dimensions, $d_{ECS}$—extracellular space thickness, $d_{fascia}$—fascia thickness, $d_{para}$—size of the parathyroid gland.

Parameter	Distribution	Distribution Indices	Value	Reference
Microscale				
$x_{cell}$ [µm]	Normal	Mean Standard deviation	7.59 µm 1.45 µm	Histology measurements [23]
$y_{cell}$ [µm]	Normal	Mean Standard deviation	7.59 µm 1.45 µm	Histology measurements [23]
$d_{ECS}$ [µm]	Uniform	Min Max	0.4 µm 0.9 µm	Estimated based on previous simulation results and on cervical epithelium measurements [24,25]
Macroscale				
$d_{fascia}$ [mm]	Uniform	Min Max	0.025 mm 0.500 mm	Estimated
$d_{para}$ [mm]	Uniform	Min Max	3 mm 8 mm	Estimated based on literature [26]

**Table 3 sensors-24-02198-t003:** The results of the global sensitivity analysis on the macroscale of the thyroid tissue multiscale model: $x_{cell}$—cell length, $d_{ECS}$—extracellular space thickness, $d_{follicle}$—size of follicle, $d_{fascia}$—fascia thickness.

	Output Parameter	Significant Input Parameter	PRCC	Correlation Level
Model including fascia	$Z_{1}$	$d_{ECS}$	−0.68	Medium
		$d_{follicle}$	−0.61	Medium
		$x_{cell}$	0.41	Medium
	$Z_{14}$	$d_{fascia}$	−0.97	High
	$f_{mid}$	$d_{ECS}$	0.73	High
		$d_{follicle}$	0.41	Medium
Model excluding fascia	$Z_{1}$	$x_{cell}$	0.39	Low
		$d_{ECS}$	−0.87	High
		$d_{follicle}$	−0.70	High
	$Z_{14}$	$d_{follicle}$	0.85	High
	$f_{mid}$	$x_{cell}$	−0.50	Medium
		$d_{ECS}$	0.87	High

**Table 4 sensors-24-02198-t004:** The results of the global sensitivity analysis on the macroscale of the parathyroid tissue multiscale model: $x_{cell}$, $y_{cell}$—cell dimensions, $d_{ECS}$—extracellular space thickness, $d_{fascia}$—fascia thickness.

	Output Parameter	Significant Input Parameter	PRCC	Correlation Level
Model including fascia	$Z_{1}$	$x_{cell}$	0.49	Medium
		$d_{ECS}$	−0.70	High
	$Z_{14}$	$d_{fascia}$	−0.88	High
	$f_{mid}$	$d_{ECS}$	0.42	Medium
		$x_{cell}$	−0.34	Low
		$d_{fascia}$	−0.77	High
Model excluding fascia	$Z_{1}$	$x_{cell}$	0.58	Medium
		$d_{ECS}$	−0.78	High
	$Z_{14}$	$y_{cell}$	−0.85	High
	$f_{mid}$	$y_{cell}$	−0.65	Medium
		$d_{ECS}$	0.44	Medium

**Table 5 sensors-24-02198-t005:** Area under curve scores for each investigated dataset based on selected spectra indices (impedance at 76 Hz—$Z_{1}$, impedance at 625 kHz $Z_{14}$, frequency in the middle of the dispersion—$f_{mid}$), t—thyroid, p—parathyroid.

Dataset	$Z_{1}$	$Z_{14}$	$f_{\mathrm{mid}}$
	t > p	p > t	p > t
in vivo dataset	0.604	0.719	0.862
	p > t	t > p	p > t
computed dataset —including fascia	0.721	0.527	0.644
	p > t	p > t	p > t
computed dataset —excluding fascia	0.732	0.998	0.971

**Table 6 sensors-24-02198-t006:** The comparison of the classifiers’ performance based on the AUC and accuracy values: SVM—Support Vector Machine, KNN—K-Nearest Neighbour and RFC—Random Forest Classifier.

Classifier	Computed with Fascia		Computed without Fascia	
	**AUC**	**Accuracy**	**AUC**	**Accuracy**
SVM	0.649	0.588	0.978	0.908
KNN	0.608	0.517	0.956	0.879
RFC	0.918	0.840	1.000	0.994

## Data Availability

The computed EIS spectra obtained through the global sensitivity analysis will be available at doi: 10.15131/shef.data.25498999.

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
