# Peer review of "The Use of Virtual Tissue Constructs That Include Morphological Variability to Assess the Potential of Electrical Impedance Spectroscopy to Differentiate between Thyroid and Parathyroid Tissues during Surgery"

_sensors, 2024, doi:10.3390/s24072198_

Round 1
Reviewer 1 Report
Comments and Suggestions for Authors
In general, the manuscript appears to cover an extensive choice of content and explores a promising topic. Consequently, I suggest revising it for submission to the MDPI: Sensor journal. However, there are some issues that need to be addressed, as outlined below:
1. Authors are advised to meticulously address the intricacies in depth. How does the presence of a surface fascia layer impact tissue separability, and how the removal of this layer might affect differentiation?
2. Examine the observed under-prediction of impedance beyond 100 kHz and its potential consequences. Investigate the factors contributing to this disparity and suggest potential approaches to rectify or comprehend this phenomenon.
3. Provide information on any validation steps taken to ensure the accuracy of the computed results, especially in comparison to the in vivo experimental data.
4. Authors must carefully deal with in detail. Why the Random Forest Classifier is chosen and how it compares to other classifiers?
5. Discuss any limitations or considerations in the selection of machine learning methods.
6. The authors employed "Figures" in the captions, whereas in the main text, they consistently referred to them as "Fig."
Comments on the Quality of English Language
Moderate correction of English is needed.
Reviewer 2 Report
Comments and Suggestions for Authors
It has been proposed for decades (e.g. Herman P. Schwan and co-workers) that the bio-impedance characteristics could be used to distinguish different types of tissues. However, the real-world application has been challenging due to the extreme complexity of real tissues.
In this manuscript, the authors conducted sensitivity and separability analysis of the thyroid and parathyroid tissue using impedance data, based on in vivo measurements and model simulations. Overall, this is a well-designed and presented study. It provides a quantitative analysis and reveals the significant influence of superficial fascia.
The only major question I have is that why did the authors only use the real part of the impedance data? The imaginary part contains substantial information regarding the membrane/tissue capacitive nature in the beta-dispersion range. Although the authors mentioned in the discussion that imaginary part could be used in the future, it is suggested that the authors explain in the method or discussion section that why it is sufficient and valid for this study to use the real part spectra only.
